# How Much Is Enough? A Surgical Perspective on Imaging Modalities to Estimate Function and Volume of the Future Liver Remnant before Hepatic Resection

**DOI:** 10.3390/diagnostics13172726

**Published:** 2023-08-22

**Authors:** Flavio Milana, Simone Famularo, Michele Diana, Kohei Mishima, Elisa Reitano, Hwui-Dong Cho, Ki-Hun Kim, Jacques Marescaux, Matteo Donadon, Guido Torzilli

**Affiliations:** 1Department of Biomedical Sciences, Humanitas University, Via Montalcini 4, 20090 Pieve Emanuele, MI, Italy; 2Division of Hepatobiliary and General Surgery, Department of Hepatobiliary and General Surgery, Humanitas Research Hospital-IRCCS, Humanitas University, Via Manzoni 56, 20089 Rozzano, MI, Italy; 3Division of Hepatobiliary Surgery and Liver Transplantation, Department of Surgery, Asan Medical Center, University of Ulsan College of Medicine, Seoul 05505, Republic of Korea; 4Research Institute Against Digestive Cancer (IRCAD), 67000 Strasbourg, France; 5Photonics Instrumentation for Health, iCube Laboratory, University of Strasbourg, 67000 Strasbourg, France; 6Department of General, Digestive and Endocrine Surgery, University Hospital of Strasbourg, 67200 Strasbourg, France; 7Department of Health Sciences, Università del Piemonte Orientale, 28100 Novara, NO, Italy; 8Department of General Surgery, University Maggiore Hospital, 28100 Novara, NO, Italy

**Keywords:** image-guided surgery, precision surgery, liver surgery, liver resection, post-hepatectomy liver failure

## Abstract

Liver resection is the first curative option for most hepatic primary and secondary malignancies. However, post-hepatectomy liver failure (PHLF) still represents a non-negligible postoperative complication, embodying the most frequent cause of hepatic-related mortality. In the absence of a specific treatment, the most effective way to deal with PHLF is its prevention through a careful preoperative assessment of future liver remnant (FLR) volume and function. Apart from the clinical score and classical criteria to define the safe limit of resectability, new imaging modalities have shown their ability to assist surgeons in planning the best operative strategy with a precise estimation of the FLR amount. New technologies leading to liver and tumor 3D reconstruction may guide the surgeon along the best resection planes combining the least liver parenchymal sacrifice with oncological appropriateness. Integration with imaging modalities, such as hepatobiliary scintigraphy, capable of estimating total and regional liver function, may bring about a decrease in postoperative complications. Magnetic resonance imaging with hepatobiliary contrast seems to be predominant since it simultaneously integrates hepatic function and volume information along with a precise characterization of the target malignancy.

## 1. Introduction

Liver resection is a potentially curative treatment for primary and secondary cancers [1,2]. Over the last decades, advancements in hepatobiliary surgery have led to reduced postoperative morbidity and mortality through improved perioperative management and a deeper understanding of liver anatomy and pathophysiology [3,4]. However, post-hepatectomy liver failure (PHLF) remains a major cause of postoperative liver-related mortality [5].

Given the lack of effective treatment protocols, preventing PHLF is crucial and relies on accurate preoperative patient evaluation and careful estimation of future liver remnant (FLR). FLR volume and metabolic function are key to PHLF development [6]. While efforts have been made to establish safe resection criteria, no silver bullet exists [7]. The accurate preoperative assessment of liver function throughout laboratory exams combined with several qualitative and quantitative risk scores is essential [8,9,10,11]. Additionally, evaluating FLR volume benefits from using advanced tools like computer-assisted operations, which enable three-dimensional planning [12,13]. Indeed, new scores and technologies may assist the surgeon in planning the best treatment option to reduce patient-related risks to a minimum [14,15].

This review aims to describe the available tools and techniques for better estimating FLR volume and functionality from a surgical perspective. Specifically, the focus is on imaging modalities and the latest technologies that will guide the new generations of liver surgeons.

## 2. PHLF: How Far We Have Come

### 2.1. Definition Criteria and Scoring Systems

PHLF represents the major cause of liver-related mortality [16]. It occurs when the liver cannot perform its synthetic and excretory functions following hepatic resection [11,17]. To identify and classify this condition, attention has been given to postoperative blood exams, including bilirubin levels and prothrombin time (PT), as well as clinical manifestations, such as hepatic encephalopathy and ascites. The widely used “50-50” criteria define PHLF as the presence of serum total bilirubin > 50 μmol/L (>2.9 mg/dL) and PT index < 50% on the fifth postoperative day [18]. Meeting these criteria is associated with a postoperative mortality rate of 59% compared to 1.2% in patients who do not meet them [18]. Subsequently, the International Study Group of Liver Surgery (ISGLS) attempted to establish a common definition, considering patient clinical signs, postoperative outcomes, and management [19]. A severity grading system was also proposed based on patient clinical management.

The occurrence of PHLF after major hepatectomies varies widely in the literature, ranging from 1% to nearly 40% [8,11,20,21,22]. The reported incidence is highest (around 20%) in patients with primary hepatobiliary cancers and concomitant chronic liver disease or cirrhosis [23,24].

### 2.2. Pathophysiology and Management

The development of PHLF is primarily determined by preoperative liver function. A preserved metabolic liver activity allows for a greater extent of resection in terms of parenchymal volume sacrifice. The relationship between function and volume is well understood, and insights into future liver remnant (FLR) pathophysiology have been gained over the years [25,26].

The regeneration process of remnant hepatocytes plays a crucial role in restoring adequate synthetic function. Certain conditions prior to surgery can impact postoperative growth and the underlying liver pathology, such as chronic liver infections, cholestatic conditions, and varying degrees of fibrosis [27,28]. Neoadjuvant protocols, leading to chemotherapy-associated liver injury (CALI), have also been associated with delayed liver recovery and increased postoperative morbidity and mortality [29,30]. Non-alcoholic steatohepatitis may also play a significant role, with reduced liver mass regeneration of about 60% compared to non-steatotic liver [31].

In addition to preoperative factors, intraoperative and postoperative events can hinder liver regeneration, even in the case of healthy parenchyma. Apart from surgical insults (e.g., ischemia-reperfusion injury, intraoperative bleeding), the amount of residual liver volume plays a critical role. Preoperative hepatopathies can lead per se to shrinkage of liver volume and decreased function, creating a delicate balance between patient needs and liver metabolic activity before resection [32]. The analysis of Zhou et al. demonstrated that a more advanced cirrhotic stage corresponded to a lower ratio of total liver volume (TLV) to standard liver volume (SLV), highlighting the delicate preoperative equilibrium between function and volume, which can impair postoperative regeneration [33,34]. Indeed, in cirrhotic livers, there is the risk of overestimating the functional liver volume, leading to inadequate limits of liver resection [35,36]. Techniques capable of evaluating liver function, such as estimating contrast enhancement ratio, offer a more sensitive approach for predicting postoperative outcomes [35,36,37].

There is no a specific remedy for PHLF, and its management has remained largely unchanged over the years, incorporating principles of acute-liver failure treatment with organ support therapies [7,38]. Mild cases with moderate hyperbilirubinemia and ascites can be managed with basic treatment involving aldosterone antagonists, loop diuretics, and fresh frozen plasma transfusions. On the contrary, the most severe cases require intensive care management. While there have been attempts to use artificial liver supports for detoxification and excretion functions, the results have been inconclusive [39,40,41].

## 3. PHLF Prevention

### 3.1. Clinical Scores and Treatment Algorithm: Definition and Drawbacks

Accurate preoperative evaluation of liver metabolic function is crucial in preventing PHLF. Due to the multifunctional nature of the liver, a comprehensive assessment requires the use of combined scores that can categorize patients into different risk classes with varying levels of reliability.

The Child–Turcotte–Pugh (CTP) score has been widely used [42]. It has been associated with the Barcelona Clinic Liver Cancer (BCLC) algorithm [43,44], but it has drawbacks, such as the “floor and ceiling effects”. Indeed, it cannot distinguish between patients with lower and higher grades of liver function, thus failing to capture differences within these heterogeneous groups.

The model for end-stage liver disease (MELD) was developed to overcome the ceiling effect of the CPT score [45]. It has been found to predict outcomes in cirrhotic patients undergoing liver resection [46]. However, it does not apply to patients without cirrhosis, who make up the majority of candidates for liver resection.

Makuuchi’s criteria represented the initial attempt to establish a proper and safer approach to planning liver resection [47]. This quantitative method incorporates three parameters: the presence or absence of uncontrollable ascites, bilirubin levels, and the indocyanine green retention rate at 15 min (ICG-R15). These values define risk categories, determining the extent of parenchymal sacrifice that can be tolerated. Applying Makuuchi’s decision tree in hepatic resection for hepatocellular carcinoma (HCC) resulted in 0% mortality in a consecutive monocentric series [11]. Indeed, the use of ICG-R15 in this context has been shown to overcome the “floor effect” of the CPT score, providing additional information to better discriminate among patients with mild impaired liver function [48].

### 3.2. ICG-R15: How to Overcome Its Limitations

Recently, drawbacks of ICG-R15 have been highlighted with the desire to expand the boundaries of liver resectability [26,49]. Certain conditions, including jaundice (with bilirubin that competes with ICG transportation), dehydration, heart failure, and arteriovenous shunts, can influence the results of the ICG-R15 test. Therefore, efforts have been made to achieve a more precise evaluation of ICG retention that compensates for the impact of circulating blood volume and sample condition [26]. When liver function cannot be adequately assessed using an ICG-based algorithm, alternative quantitative function assessments have been proposed, such as the Technetium-99m-labeled diethylenetriamine pentaacetic acid galactosyl human serum albumin (TcGSA) scintigram [50,51], the galactose elimination capacity [52], and more recently, the hepatobiliary scintigraphy using iminodiacetic acid (IDA) derivatives [53]. However, none of those assessments have gained a prominent role in diagnostic and therapeutic algorithms.

Furthermore, the ICG-R15 test has been criticized for grouping a wide range of patients with varying liver functional reserves (e.g., ICG-R15 values of 20% and 29% being classified similarly). To address these controversies, an expansion of Makuuchi’s criteria has been attempted, with a specific focus on FLR volume [6,26]. Besides, some reports have argued that these criteria demonstrate high sensitivity but low specificity in predicting PHLF, and no significant correlations have been observed between the decision tree and the occurrence of grade B or C PHLF [26,54]. In this evolving landscape, precise estimation of FLR volume has gained increasing importance not only for primary but also for secondary liver tumors in the new era of parenchymal-sparing resection [4,55], often complicated by CALI [54].

## 4. PHLF Prevention: The Role of Imaging Techniques

### 4.1. The Importance of Remnant Liver Volume Percentage

From the deepest understanding of how Makuuchi’s criteria have influenced the safety and effectiveness of liver resection, the importance of FLR percentage volume has emerged. Takasaki and colleagues were the first to highlight the utility of the FLR plasma clearance rate of ICG in estimating FLR function [56]. On this concept, Nagino et al. [6,57] proposed that ICGK-rem < 0.05 (preoperative ICG-R15 multiplied by the percentage of remnant liver volume) could be the strongest predictor of postoperative mortality in biliary cancer resection. Given the prognostic significance of resectability in a patient’s oncological history, efforts have been made to further push this cutoff value, aiming to expand the range of resectable diseases.

Originally, Makuuchi recommended portal vein embolization (PVE) when the anticipated FLR volume was less than 40% in patients with a normal liver and 50% in patients with impaired liver function as defined by ICGR-15 [3,58]. Over time, the authors introduced additional factors, such as bilirubin levels [59], cholinesterases [10], and liver stiffness estimation, to refine the cutoff value [60]. Consequently, the FLR threshold was further reduced. In the Western world, Vauthey et al. suggested that a resection could be considered safe with an FLR of >20% for a healthy liver, >30% in case of fibrosis, and >40% in the presence of cirrhosis [61,62,63]. Conversely, Kokudo et al. still considered >50% as the safe limit for patients with cirrhotic livers and an ICG-R15 < 14% as the cutoff for major hepatectomies without PVE [14]. More recently, Truant et al. proposed that the remnant liver volume to body weight ratio (RLV/BWR) was more specific than the traditional remnant liver volume/total liver volume ratio (RLV/TLV) for non-cirrhotic livers, suggesting that RLV/BWR < 0.5% could serve as a new cutoff for estimating postoperative hepatic dysfunction [64].

However, it is important to consider that the minimum FLR depends on the institutional policy regarding acceptable mortality rates. Even if a low FLR volume may theoretically permit a safe liver resection, postoperative complications such as sepsis can significantly impact patient survival when a minimal FLR volume cannot adequately respond to external insults [65].

### 4.2. The Guidance of 3D Volume Reconstruction

In assessing liver remnants, apart from volume estimation, 3D reconstruction software has opened up possibilities for reconstructing vascular territories and visualizing tumoral dissemination, enabling virtual liver resection. This approach allows for a precise evaluation of the impact of surgery on the FLR, including the appropriate management of vascular inflows and outflows [13,15].

The concept of liver 3D rendering was first introduced by Hashimoto et al. in the 1990s, demonstrating the manual reconstruction of liver structures’ contours [66]. Marescaux later described the initial experience with automated 3D modeling in 1998 [67]. Subsequently, the volumetric analysis based on 3D reconstruction became established in living donor liver transplantation (LDLT) and liver resection [68,69]. The use of 3D reconstruction for operation planning was first reported in 2000 [70], and the first hepatectomy guided by real patient hepatic blood flow circulation was documented in 2005 [71]. This advancement allowed for a precise estimation of FLR volume, facilitating the prediction of potential complications such as venous congestion and parenchyma sacrifice. It overcame the limitations of 2D volumetry, which relied on manual tracing of volume along anatomical landmarks and was limited to the four sections of the liver [71,72]. Surgeons have progressed from applying this system to basic procedures to embracing new paradigms, using 3D simulation software in patients with multiple bilobar colorectal liver metastases (Figure 1) [13].

In such cases, the complexity lies not only in establishing a safe liver resection cutoff but also in accurately predicting FLR along intricate surgical planes. Traditional 2D hand-traced techniques could not precisely estimate the percentage of FLR volume, whereas numerous successful experiences have demonstrated the value of 3D modeling with specialized software capable of reproducing hepatic structures from computed tomography (CT) scans or magnetic resonance imaging (MRI). Several studies have supported the role of 3D simulation by showing a strong correlation between the actual and the predicted volume of resected liver specimens, enabling tailored liver resection on an individual basis.

However, the utility of 3D modeling extends beyond preoperative feasibility assessment; it also serves as a didactic tool, enhancing anatomical and surgical understanding for the new generation of surgeons [13]. Majno et al. emphasized that for precise liver surgery, it is essential to comprehend the exact anatomical configuration at a deeper level rather than relying solely on the “1-2-20 concept”, and 3D reconstruction plays a critical role in this process [73]. Furthermore, the benefits of 3D preoperative simulation have been observed as an intraoperative advantage, with shorter operative times in patients who underwent preoperative 3D reconstruction than those who had not. These effects were particularly prominent in cases of repeat hepatectomy when the liver anatomy may not be as clear [74].

### 4.3. Hepatobiliary Scintigraphy and SPECT/CT

Traditionally, the determination of safe percentages for FLR volume has varied based on the presence or absence of underlying liver disease. However, estimating FLR metabolic activity has proven challenging, necessitating the development of imaging techniques for assessing remnant function in recent years.

In this context, dynamic hepatobiliary scintigraphy (HBS) has emerged as a valuable modality for quantitatively evaluating total and regional liver function. Among the commonly used technique and radiopharmaceutical tracers, Technetium-99m(99mTc)-labeled diethylenetriaminepentaacetic acid (DTPA) galactosyl human serum albumin (GSA) scintigraphy and hepatobiliary scintigraphy (HBS) with 99mTc-labeled iminodiacetic acid (IDA) derivatives are prominent examples [75]. Although both methods provide quantitative and visual information regarding hepatic function, 99mTc-mebrofenin represents the most hepatic-specific 99mTc-IDA derivative and is commonly employed [53]. Mebrofenin, similar to bilirubin, is taken up by hepatocytes and excreted into bile canaliculi without undergoing biotransformation. As a result, it has been utilized to predict PHLF in patients undergoing liver resection [76,77].

De Graaf et al. demonstrated the correlation between liver function and remnant volume using HBS, particularly in the case of normal liver function, whereas patients with underlying hepatopathy exhibited lower liver functionality per residual volume [76]. Based on the uptake rate of 99mTC-mebrofenin (expressed as %/min of injected dose) and body surface area (BSA), a cutoff was initially established, namely, FLR-Function (FLR-F = 2.69%/min/m^2^), to identify patients at risk of developing PHLF in the reported series.

One advantage of HBS over previously reported clinical score systems is its ability to focus on specific regions of interest (ROI), revealing that parenchymal damage may not be evenly distributed, as reported in pathological studies [78]. Consequently, the prevention of PHLF is not solely dependent on volume considerations.

Notably, HBS has also been employed to assess biliary drainage, as meborfenin’s biliary excretion allows for the evaluation of cholestatic conditions affecting the entire liver, as well as segmental organ dysfunction [79]. This enables a more precise assessment of cholestatic patients than relying solely on serum bilirubin levels, even in cases where preoperative drainage is necessary, such as in patients with peri-hilar cholangiocarcinoma. Additionally, the ability of HBS to predict the functionality of specific liver areas has proven valuable in evaluating the efficacy of hypertrophic techniques like PVE and Associating Liver Partition and Portal vein ligation for Staged hepatectomy (ALPPS) [80] other than living donor liver transplantation [81].

Moreover, the recent advancements brought about by the dual camera and the three-dimensional visualization of the SPECT/CT have further improved measurement accuracy, expanding the applicability of HBS. A deeper understanding of the relationship between function and volume using HBS has revealed that the increase in liver volume after an ALPPS procedure does not always correspond to a proportional increase in liver function [82]. With various definitions of PHLF adopted in recent decades, different FLR-F cutoff values have been proposed [76,77,83]. Furthermore, a recent development in assessing remnant liver function is the Hospital Italiano de Buenos Aires (HIBA) index, proposed as an alternative measurement [84]. Recent reports have indicated that instead of relying on a specific cutoff value, considering a range with upper and lower limits is more appropriate for accurately predicting PHLF in accordance with the latest definition by the ISGLS [85].

### 4.4. Combining Volume and Function: The Role of MRI

When considering the emerging methods for predicting PHLF, it becomes evident that a technique capable of accurately estimating both the volume and function of the FLR is crucial. In this context, MRI has gained prominence due to its ability to provide comprehensive information about primary or secondary liver tumors, as well as specific anatomical and quantitative functional information for the entire liver and specific regions [86,87,88,89]. In this respect, MRI offers the advantage of predicting FLR regeneration, thereby further reducing the risk of PHLF [90,91]. Notably, the use of the paramagnetic hepatobiliary contrast agent Gadolinium ethoxybenzyl- diethylenetriaminepentaacetic acid (Gd-EOB-DTPA) in T1-weighted imaging has demonstrated superior reliability in reflecting liver function and staging fibrosis compared to non-contrast enhanced DWI-phases, as shown by Watanabe et al. [92]. Unlike other extracellular agents, approximately 50% of the administered Gd-EOB-DTPA is actively taken up by hepatocytes [3,4] and excreted into the bile without being metabolized [93]. The resulting circulating Gd-EOB-DTPA exhibits a T1-shortening effect that is proportional to its concentration in the blood during the portal phase to the concentration in liver tissue during the hepatobiliary phase [94,95]. The enhancement of liver parenchyma results from an interplay between the uptake activity of organic anion-transporting polypeptides and the elimination process mediated by multidrug resistance protein 2 transporters into the bile duct. The expression of these transporting polypeptides has been inversely correlated with the severity of liver disease [93,96].

Given these dynamics, Gd-EOB-DTPA contrast-enhanced MRI can potentially estimate hepatic flow and function, which are progressively affected in hepatic damage conditions.

#### 4.4.1. MRI Applicability in Estimating Liver Function and Volume

To address this objective, various strategies have been proposed to extract meaningful data from MRI sequences that are closely associated with the liver substrate (Appendix A). Notably, Nilsson et al. demonstrated the relationship between MRI findings and hepatic diseases such as primary biliary cirrhosis, providing valuable insight into liver pathology [97]. Similarly, Kim et al. reported a significant correlation between the degree of Gd-EOB-DTPA enhancement and liver function parameters, highlighting the diagnostic potential of MRI in assessing liver function [98].

Comparisons have been made between MRI-derived ratios and established liver function scores, including the CTP and MELD scores. The contrast-enhancement ratio of the liver in the hepatobiliary phase (CERH) was found to significantly decrease in CTP A and B patients [37]. Other studies showed that signal intensity (SI) in the hepatospecific phase (SI-HEP) could distinguish statistically significant differences between low (<10) and high (>10) MELD cases, effectively identifying patients with decompensated cirrhosis [99,100]. Ippolito et al. demonstrated, for the first time in a large retrospective cohort, the possibility of effectively stratifying patients with different CTP and MELD scores based on SI-HEP measurements [88].

Regarding liver function assessment, the contrast enhancement ratio (CER) between transitional and hepatobiliary phases outperformed other commonly used estimators, such as TLV/SLV [35].

Furthermore, novel indices of liver function, including relative enhancement (RE) and contrast enhancement ratio (CER), derived from the comparison of SI measurement and apparent diffusion coefficient (ADC) maps in precontrast and Gd-EOB-DTPA T1-weighted images and hepatobiliary phases, have been associated with the development of PHLF [35]. While RE was more influenced by hemodynamics (extracellular contrast), CER emerged as an independent predictor of PHLF, with remnant liver CER demonstrating superior diagnostic performance compared to RLV/SLV in distinguishing PHLF from non-PHLF cases [35].

In addition, Yoneyama et al. reported significant correlations between Gd-EOB-DTPA-enhanced MRI and ICG clearance, potentially overcoming the drawbacks of the ICG-R15 test [101]. However, a consensus has not been reached regarding the optimal MRI estimator of liver function, which despite the widespread use of SI-derived parameters, remains the most widely used other than the simplest one [102,103]. In many cases, these scores represent different mathematical expressions of the same contrast intrahepatic flow/uptake phenomenon. Recently, Wang et al. evaluated the correlation between liver function of 12 frequently used parameters derived from SI-analysis and T1-relaxometry in groups with and without liver disease [104]. They found that simple SI-based parameters, despite some technical limitations, were as effective as the more complex ones in evaluating liver function.

#### 4.4.2. The Current Role of MRI and Future Perspectives

Despite the impressive potential of Gd-EOB-DTPA-enhanced MRI, it is important to note that there are currently no published studies directly comparing this imaging modality with other described techniques. Therefore, detailed analyses of these techniques have not been reported. However, based on the available evidence and setting aside the discussion on the best estimator score, it is evident that MRI, which provides both functional and volumetric liver information, may represent the most comprehensive preoperative assessment tool. Consequently, its use in preoperative risk assessment is expected to increase, potentially playing a key role in the strategy for preventing PHLF (Figure 2).

## 5. Conclusions

To summarize, PHLF is a significant cause of liver-related morbidity and mortality, with no effective treatments besides supportive therapies. Therefore, precise preoperative estimation of the liver remnant is crucial in reducing the risk of PHLF. Although various scoring systems exist, each has limitations, resulting in disparities among surgical groups.

In this context, emerging technologies such as 3D volume reconstruction offer reliable tools for planning safe hepatic resection. However, there is still a lack of consensus regarding volume cutoff points. HBS provides valuable functional information, while MRI enables simultaneous characterization of patient-specific oncological disease and liver metabolic substrate, making it a highly useful tool in the preoperative assessment algorithm. MRI images not only serve as a basis for volumetric 3D reconstruction to facilitate hepatectomy planning but also allow for the estimation of liver functional reserve. Thus, MRI can potentially address several unanswered questions when dealing with liver resection. In light of this, among the array of suggested technologies, MRI may represent the most promising diagnostic tool. However, the integration of algorithms incorporating Gd-EOB-DTPA-enhanced MRI for assessing metabolic function in routine preoperative evaluations remains to be standardized. This issue could potentially be addressed through upcoming collaborative multicenter face-to-face studies. Such studies have the potential to validate and acknowledge the substantial role of MRI in preventing PHLF in comparison to other imaging techniques.

## Figures and Tables

**Figure 1 diagnostics-13-02726-f001:**
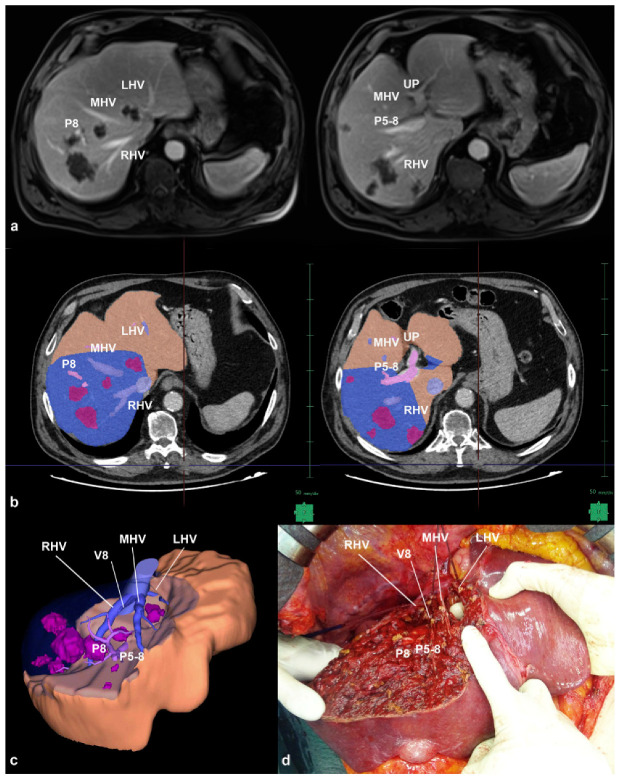
(**a**) MRI-scan of a patient with multiple bilobar and deep-located colorectal liver metastases in contact with all three hepatic veins at their caval confluence, the portal pedicle to S8 (P8) at its origin from P5-8; (**b**) manual tracing of the outline of resection area (blue area) on each CT section to obtain the 3D virtual cast; (**c**) 3D virtual cast showing the planned hepatectomy; (**d**) the hepatectomy consisted in a wide partial resection of segments 4 superior, 8 and 5, enlarged to the segment 6, segment 7, the paracaval portion of segment 1; all three hepatic veins, P8 and P5-8 were fully exposed on the cut surface; predicted future liver remnant was 994 mL while the real was 1006 mL (error rate −0.7%) (adapted from [13] with permission).

**Figure 2 diagnostics-13-02726-f002:**
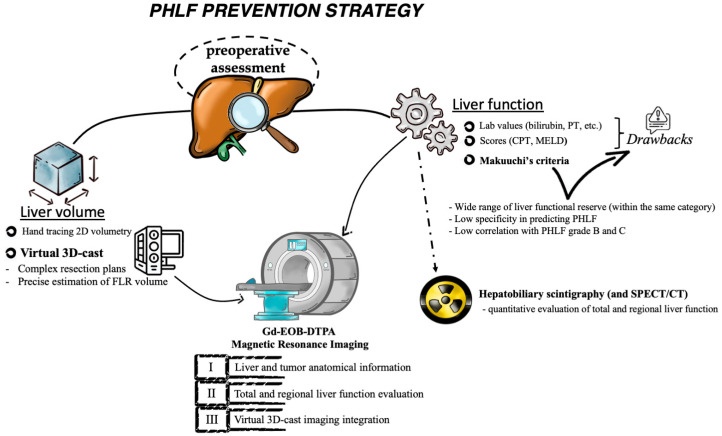
Diagnostic tools to assess the liver remnant function and volume with the possibility to guide the resection plan to reduce the risk of developing PHLF.

## Data Availability

Not applicable.

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
