# Peer review of "How Much Is Enough? A Surgical Perspective on Imaging Modalities to Estimate Function and Volume of the Future Liver Remnant before Hepatic Resection"

_diagnostics, 2023, doi:10.3390/diagnostics13172726_

Round 1

Reviewer 1 Report (Previous Reviewer 1)

The authors have addressed the comments of the reviewers, and the revised version is being suggested for publication in this journal.

.

Author Response

We would like to express our sincere gratitude for the review of our paper. Your insightful feedback and thorough examination have been truly invaluable. We appreciate your time and effort in ensuring the quality of the manuscript. Your constructive insights have undoubtedly contributed to its refinement.

Reviewer 2 Report (Previous Reviewer 2)

The authors have clearly improved their manuscript which is currently of full scientific soundness. I understand the meaning of Figure 1, but I am missing an example of MRI based FLR: this would increase the visibility of the paper, at publication but also further on web platforms!

Last paragraph of the conclusion: ‘MRI represents the most promising…’ should better read ‘MRI may represent the most promising…’, in line with the abstract. I would suggest the authors to call for face-to-face multicentric studies to compare their proposed MRI methodology with other techniques. This can lead to validation of a working algorithm, such as proposed in Fig. 2.

I propose a list of minor changes, mainly textual:

-          Abstract, last line: ‘other than’ should better read ‘along with’

-          Page 3, section 3.1: the authors should refer to the Child-Turcotte-Pugh (CTP) and not to the Child-Pugh-Turcotte (CPT) score, even though Child-Pugh is the most commonly used denomination

-          Page 3 section 3.1: Although obvious, I am not sure that BCLC was previously defined

-          Page 7, last line: meter square is spelt m^2, should be m²

-          Page 8: I am not sure ALPPS and HIBA have been defined as abbreviations

Page 9, last line title of section 4.4.2: 'actual' should read ‘current’.

Minor editing efforts

Author Response

To begin, we express our sincere gratitude to the reviewer for the insightful suggestions, which have significantly contributed to enhancing the quality of the manuscript. We have meticulously addressed each comment in a point-by-point manner and subsequently made the necessary revisions to the manuscript (highlighted in red). We are hopeful that these revisions align the manuscript more effectively with the standards required for publication.

The authors have clearly improved their manuscript which is currently of full scientific soundness. I understand the meaning of Figure 1, but I am missing an example of MRI based FLR: this would increase the visibility of the paper, at publication but also further on web platforms!

We thank the reviewer for the valuable suggestion. The authors firmly acknowledge the pivotal role of MRI in this context. To emphasize this significance, we have thoughtfully incorporated a new Figure 1 into the manuscript, hoping the illustrative addition better delineates the role of MRI.

Last paragraph of the conclusion: ‘MRI represents the most promising…’ should better read ‘MRI may represent the most promising…’, in line with the abstract. I would suggest the authors to call for face-to-face multicentric studies to compare their proposed MRI methodology with other techniques. This can lead to validation of a working algorithm, such as proposed in Fig. 2.

We thank the reviewer for the comment, which prompted us to revise the text to advocate for face-to-face studies that can facilitate a comparative analysis between MRI and other methodologies.

I propose a list of minor changes, mainly textual:

All the minor revisions have been addressed along the text.

-          Abstract, last line: ‘other than’ should better read ‘along with’

-          Page 3, section 3.1: the authors should refer to the Child-Turcotte-Pugh (CTP) and not to the Child-Pugh-Turcotte (CPT) score, even though Child-Pugh is the most commonly used denomination

-          Page 3 section 3.1: Although obvious, I am not sure that BCLC was previously defined

-          Page 7, last line: meter square is spelt m^2, should be m²

-          Page 8: I am not sure ALPPS and HIBA have been defined as abbreviations

Page 9, last line title of section 4.4.2: 'actual' should read ‘current’.

This manuscript is a resubmission of an earlier submission. The following is a list of the peer review reports and author responses from that submission.

Round 1

Reviewer 1 Report

Post-hepatectomy liver failure (PHLF) is a potential major complication after liver resections. Different scores, imaging techniques and methods to calculate liver function aim to reduce the risk of PHLF.   A large range of methods to predict future liver remnant (FLR) volume and function are discussed in this review. The task of giving an overview of this important topic is of high interest!   Further suggestions may be given for further improvement:   The manuscript needs major editing by a native English speaker.   The text could be structured more stringently (PHLF: definitions, pathophysiology (intra/extrahepatic). Scores/imaging: definition, results, advantages, and limitations.   Some redundancies could be eliminated.   In the abstract "most cost-effective resection planes" are mentioned but no cost analyses are detailed in the manuscript.   I suggest summarizing key scores and modalities of imaging and functional analysis in a table (e.g. definition, advantage, limitation, key reference with publication year). There may be an "overlap" with Figure 2 which is acceptable.   Table 1 can be omitted or shifted to Supplements. This information is important if the authors intend to publish this review in a radiological journal.   The authors frequently limit their description to rather vague summaries such as "have a good discriminating power", show a "valid correlation" or "have a stronger diagnostic performance". Precise results could be added and would make it easier for the reader to assess the advantages and limitations of the discussed scores/imaging modalities.

Author Response

The authors really appreciated the comments. 

Post-hepatectomy liver failure (PHLF) is a potential major complication after liver resections. Different scores, imaging techniques and methods to calculate liver function aim to reduce the risk of PHLF.   A large range of methods to predict future liver remnant (FLR) volume and function are discussed in this review. The task of giving an overview of this important topic is of high interest!   

The authors do appreciate very much the time the reviewer spent in reading the presented paper. We believe the received comments may give our work an increased value and appropriate modification to the text has been made and highlighted in red. A reply point by point has been provided as well.

Further suggestions may be given for further improvement:   

  • The manuscript needs major editing by a native English speaker.

Thank you for the suggestion. The manuscript underwent an extensive English revision by appropriate service.

  • The text could be structured more stringently (PHLF: definitions, pathophysiology (intra/extrahepatic). Scores/imaging: definition, results, advantages, and limitations.  Some redundancies could be eliminated.  
    Thank you for your comment. As suggested, a better structure of the text has been proposed in particular for the first part. Some redundancies have been eliminated along the text and the introduction has been reduced as well.

  • In the abstract "most cost-effective resection planes" are mentioned but no cost analyses are detailed in the manuscript.
    Thank you for the comment. The main intent of the review was to analyze PHLF as a major post-operative complication from a clinical point of view. That “cost-effective” in the abstract section was not related to economical aspects. On the contrary, it was referred to the ability of the surgeon to choose a resection plane that can save as much liver parenchyma as possible. Indeed, its metaphoric meaning would compare the cost of liver parenchymal percentage sacrifice to the effect of an oncological appropriate resection. However, we agree that a more specific explanation was needed as modified in the text.

  • I suggest summarizing key scores and modalities of imaging and functional analysis in a table (e.g. definition, advantage, limitation, key reference with publication year). There may be an "overlap" with Figure 2 which is acceptable.  
    Thank you for the interesting suggestion. An appropriate table has been created and added to the text (table 1)

  • Table 1 can be omitted or shifted to Supplements.
    Thank you for your suggestion. The previous Tab.1 was shifted to Supplements.

  • This information is important if the authors intend to publish this review in a radiological journal. 
    We considered the comment referring to the creation of a new table summarizing all the aspects of our review as we did for the “new” table 1.

  • The authors frequently limit their description to rather vague summaries such as "have a good discriminating power", show a "valid correlation" or "have a stronger diagnostic performance". Precise results could be added and would make it easier for the reader to assess the advantages and limitations of the discussed scores/imaging modalities.
    Thank you for the comment. The exact statistic behind the cited studies was not reported deliberately since the aim of the work was to review the current state of literature in a broad sense. However, we agree with the reviewer that some phrases may sound too informal and for this reason we tried to modify the text, in particular the MRI section, to sound more scientific.

Reviewer 2 Report

The authors present a comprensive review of the importance of identifying risk factors for post-hepatectomy liver failure. The introduction is far too long and should go straight to the point. The goal is clearly to promote the use of hepatobiliary-enhanced contrast MRI as the most promising technique. The authors do not show any convincing evidence that this technique is better than any other (i.e. ICG clearance combined with CT volumes or 99mTc-HIDA scan).

The authors should reformat their paper in a demonstration of the potential of MRI to help in this, since, to my knowledge, no comparative, face-to-face comparisons exist: from any alledged data can be demonstrated that MRI is the best choice.

The paper is however important and should be revised accordingly, but presented in this way, it is laudatory and not scientifically sound.

Other comments:

Figure 1 is an extreme case that most surgeons will rarely see: I do not see this as a good example of the problematic: further it does not support in any way the statements as no MRI data are provided.

References 58-59 refer to 99mTc-GSA, not to 99mTC-DTPA-GSA radiotracers... Please check what is meant!

Nothing or very little is mentioned about portal vein thrombosis, which is usually a contraindication to surgery: the authors should refer to it in the introduction because it is a kind of no-go for reductive liver surgery

To be honnest, I am not in the position to assess the validity of the MRI evaluation methods and I hope that another reviewer may have an opinion on this. 

Author Response

The authors really appreciated the comments. 

We have really appreciated the comments and suggestions the reviewer gave us to improve our paper. Following the points highlighted by the review process an appropriate modification to the text has been carried out and tracked in red. A reply point by point has been provided as well.

  • The authors present a comprensive review of the importance of identifying risk factors for post-hepatectomy liver failure. The introduction is far too long and should go straight to the point.
    Thank you for the comment. The length of the introduction has been reduced and the text has been structured more stringently in order to eliminate some redundancies.

  • The goal is clearly to promote the use of hepatobiliary-enhanced contrast MRI as the most promising technique. The authors do not show any convincing evidence that this technique is better than any other (i.e. ICG clearance combined with CT volumes or 99mTc-HIDA scan).
    We really appreciate your comment that actually highlights a weakness of our paper. At our knowledge in this moment there are now published studies comparing directly the results of the several techniques in order to show a clear superiority of one over the others. Our point simply derives from a list of pros and cons of the reported imaging modalities. However, a final statement has been added to clarify the lack of convincing evidence.

  • The authors should reformat their paper in a demonstration of the potential of MRI to help in this, since, to my knowledge, no comparative, face-to-face comparisons exist: from any alledged data can be demonstrated that MRI is the best choice.
    Thank you for the comment. We hope that from the added table and the more scientific soundness given to the paper, the role of MRI can be more easily extrapolated. Since no comparative data exist between the advantage of MRI and the other described techniques, clear conclusions cannot be extrapolated, and this limitation has been stated in the text.

The paper is however important and should be revised accordingly, but presented in this way, it is laudatory and not scientifically sound.
The authors really thank the reviewer for the comment. We re-elaborated the paper according to the given suggestion hoping our work could now sound more detailed and scientific.

Other comments:

  • Figure 1 is an extreme case that most surgeons will rarely see: I do not see this as a good example of the problematic: further it does not support in any way the statements as no MRI data are provided.
    Thank you for your comment. We understand that the figure proposed may represent an extreme case for most of the HPB surgeons. However, we do believe that the disease presentation of bilobar colorectal liver metastases (CLM) is the best example to explain the importance of a careful preoperative evaluation in order to reduce any case of PHLF related to insufficient liver volume remnant. Moreover, the concept of one-stage hepatectomy has been largely detailed in literature and even more complex scenario have been described. The provided case refers to a reconstruction made on CT-scan as at that point in the text the authors would focus on the importance of liver volume and not function. Indeed, the crucial role of MRI lies in the capacity of resuming both volume and function information other than the possibility to integrate with 3D reconstruction software. All these peculiarities have been explicated in the MRI chapter.

  • References 58-59 refer to 99mTc-GSA, not to 99mTC-DTPA-GSA radiotracers... Please check what is meant!

Thank you for your comment. At that point the authors would underline the presence of alternatives to ICG-R15 in assessment of hepatic function, more than hepatic flow, as described in the cited papers. that refer to 99mTc-GSA and galactose elimination capacity (GEC). We agree with the reviewer that the text could be misleading and for this reason it has been adjusted.

  • Nothing or very little is mentioned about portal vein thrombosis, which is usually a contraindication to surgery: the authors should refer to it in the introduction because it is a kind of no-go for reductive liver surgery
    The authors really thank the reviewer for pointing out another fundamental concept in liver resection as the portal vein thrombosis. Indeed, it may represent a contraindication to resection when considering some treatment algorithms.
    However, the aim of the review was to focus on preoperative assessment of liver remnant analyzing the available imaging techniques. We do believe that opening the debate also to the many existing contraindication for liver resection would inevitably stray off topic.

To be honnest, I am not in the position to assess the validity of the MRI evaluation methods and I hope that another reviewer may have an opinion on this. 

We really appreciated the comment and the honesty of the reviewer, we hope that modification we made to our paper would accomplish what suggested.

Round 2

Reviewer 2 Report

The paper has been well improved, and thanks to the authors for this;

The only flaw remains with the radiopharmaceuticals to be used fot assessing the lobar/segmental liver function together with CT (SPECT/CT) or coregistration with MRI. Nowadays, (dis)IDA derivatives are the most used and shoud be cited instead of the old fanshioned DTPA-HSA-galactosyl derivatives.

All of them can be cited, but not the last forgotten.

Author Response

The authors really appreciated the positive comment of the reviewer. We modified the manuscript according to the last suggestion given. Appropriate citations have been added and all new modification can be found in green